# Assessing Huanglongbing Severity and Canopy Parameters of the Huanglongbing-Affected Citrus in Texas Using Unmanned Aerial System-Based Remote Sensing and Machine Learning

**DOI:** 10.3390/s24237646

**Published:** 2024-11-29

**Authors:** Ittipon Khuimphukhieo, Jose Carlos Chavez, Chuanyu Yang, Lakshmi Akhijith Pasupuleti, Ismail Olaniyi, Veronica Ancona, Kranthi K. Mandadi, Jinha Jung, Juan Enciso

**Affiliations:** 1Texas A&M AgriLife Research & Extension Center, 2415 E. Highway 83, Weslaco, TX 78596, USA; ittipon.kh@tamu.edu (I.K.); jose.chavezortiz@ag.tamu.edu (J.C.C.); lakshmi.pasupuleti@ag.tamu.edu (L.A.P.); kkmandadi@email.tamu.edu (K.K.M.); 2Department of Plant Production Technology, Faculty of Agricultural Technology, Kalasin University, Kalasin 46000, Thailand; 3Department of Biological and Agricultural Engineering, Texas A&M University, 2132 TAMU, College Station, TX 77843, USA; 4Department of Agriculture, Agribusiness and Environmental Sciences, Texas A&M University-Kingsville, Citrus Center, 312 N. International Blvd., Weslaco, TX 78596, USA; chuanyu.yang@tamuk.edu (C.Y.); veronica.ancona-contreras@tamuk.edu (V.A.); 5Lyles School of Civil Engineering, Purdue University, West Lafayette, IN 47907, USA; iolaniyi@purdue.edu (I.O.); jinha@purdue.edu (J.J.); 6Department of Plant Pathology and Microbiology, Texas A&M University, 2132 TAMU, College Station, TX 77843, USA; 7Institute for Advancing Health Through Agriculture, Texas A&M AgriLife, College Station, TX 77843, USA

**Keywords:** greening disease, antimicrobials, vegetation indices, texture features

## Abstract

Huanglongbing (HLB), also known as citrus greening disease, is a devastating disease of citrus. However, there is no known cure so far. Recently, under Section 24(c) of the Federal Insecticide, Fungicide, and Rodenticide Act (FIFRA), a special local need label was approved that allows the trunk injection of antimicrobials such as oxytetracycline (OTC) for HLB management in Florida. The objectives of this study were to use UAS-based remote sensing to assess the effectiveness of OTC on the HLB-affected citrus trees in Texas and to differentiate the levels of HLB severity and canopy health. We also leveraged UAS-based features, along with machine learning, for HLB severity classification. The results show that UAS-based vegetation indices (VIs) were not sufficiently able to differentiate the effects of OTC treatments of HLB-affected citrus in Texas. Yet, several UAS-based features were able to determine the severity levels of HLB and canopy parameters. Among several UAS-based features, the red-edge chlorophyll index (CI) was outstanding in distinguishing HLB severity levels and canopy color, while canopy cover (CC) was the best indicator in recognizing the different levels of canopy density. For HLB severity classification, a fusion of VIs and textural features (TFs) showed the highest accuracy for all models. Furthermore, random forest and eXtreme gradient boosting were promising algorithms in classifying the levels of HLB severity. Our results highlight the potential of using UAS-based features in assessing the severity of HLB-affected citrus.

## 1. Introduction

Citrus is one of the important fruit crops in the U.S., primarily grown in Florida, California, and Texas. According to the United States Department of Agriculture, 7.78 million tons of citrus were produced in 2020, with an economic value of approximately USD 3 billion [1]. In recent years, however, the U.S. citrus industry has encountered major production losses due to huanglongbing (HLB), also known as greening disease [2]. A yield reduction of 76% affected by HLB has been reported in Florida [3]. HLB, a devastating disease of citrus, is associated with the phloem colonizing bacterium Candidatus Liberibacter asiaticus (CLas) [4] in the U.S. and is spread by Asian citrus psyllid [5]. So far, there is no available cure for HLB, even though extensive scientific investigations are being pursued [6].

The following are some of the symptoms of HLB-affected citrus, for example, mottling or blotchy mottle, chlorosis, thinner canopy, and sparse foliation. In Florida, HLB has spread to all growing areas, and there are no uninfected control trees [7]. Recently, a special local need label was approved that allows the trunk injection of antimicrobials such as oxytetracycline (OTC) for HLB management in Florida [8]. OTC improved the fruit yield, quality, and tree health of HLB-affected trees in Florida [9]. In Texas, although HLB is considered endemic, the disease pressure and rate of spread are much lower than in Florida [10], and the impact on yields is relatively lower than in Florida, possibly due to differences in production practices, cultivars, and environmental factors [10]. Still, there is a need for the assessment of HLB disease severity, which is essential for HLB control, minimizing yield loss, and evaluating HLB-tolerant cultivars in addition to investigating the impact of potential new antimicrobials in Texas citrus-producing regions. The importance of a reliable assessment of HLB severity is not only helpful in evaluating the HLB-affected citrus after treatments are applied but also could help growers’ decision-making in the long term. The typical method to determine HLB incidence or severity is by quantifying CLas using polymerase chain reaction (PCR). This method is used to assess if a treatment could reduce the level of CLas population in the HLB-affected citrus trees [7]. In the field, HLB severity is assessed by a Disease Index Rating [11], but this subjective rating method, however, requires a well-trained technician to obtain a reliable assessment.

In addition, the most important and relevant value for evaluating HLB treatments for growers is the effect on the fruit yield of the tree, but yield measurement is very labor-intensive and can only be measured during harvesting time, which is impossible to measure during the juvenile stage [7]. Consequently, many have attempted to evaluate tree health parameters (e.g., canopy volume, canopy density, canopy color) to assess the health status of citrus [12,13]. Recently, Levy et al. [7] reported that canopy density was an effective trait in predicting the yield of HLB-affected trees, and it is more relevant than the CLas titers.

Due to its visible symptoms, many scientists have been able to detect the HLB-affected citrus tree using UAS-based remote sensing techniques [14,15,16]. According to the existing literature review, several articles have reported using remote sensing tools, together with machine learning, in classifying plant disease severity such as leaf blast in rice [17,18] and tar spot disease in corn [19,20]. However, very few, if any, have aimed to classify the severity of HLB-affected citrus trees using such technology.

The most common remote sensing data are spectral information, which relies on leaf reflectance, such as vegetation indices (VIs); moreover, they have successfully been used to detect HLB-affected citrus trees [13,14,16,21]. Textural features (TFs), on the other hand, provide spatial arrangement information of the color or intensity in an image [22]. It has been widely used for image classification purposes and for biomass estimation [23]. The combination of spectral data with texture features, therefore, captures new variations that could not be captured using only either one, so it often improves the accuracy of the phenotyping task. Therefore, the objectives of this study were (1) to use UAS-based remote sensing techniques to assess the effectiveness of OTC on the HLB-affected citrus in Texas; (2) to use UAS-based remote sensing to distinguish the levels of HLB severity and canopy health; (3) to leverage UAS-based features, along with machine learning, for HLB severity classification; and (4) to assess the effect of different flight altitudes on VIs and TFs.

## 2. Materials and Methods

### 2.1. Study Area and Experimental Design

The study was carried out at Texas A&M University–Kingsville Research farm, located in Weslaco, Hidalgo County, Texas. The study area is shown in Figure 1. The experimental design was a randomized complete block design (RCBD), including 3 treatments and 3 blocks. The treatments were performed by trunk injection of ArborOTC (0.7 g/tree, T1) in ~10-year-old HLB-affected citrus trees (var. Rio Red grapefruit) alongside water (T2) and mock (T3) controls.

### 2.2. Visual Assessment of HLB Severity and Canopy Parameters

The treatments were applied in February 2023. Afterward, visual ratings of HLB severity, canopy color, and canopy density were recorded monthly. We used the criteria previously reported by Archer et al. [24] to assess the visual rating. HLB symptoms that were considered included leaf chlorosis, blotchy mottle, and corky veins. These assessments were accomplished by a well-trained scientist. The categories of the assessment are shown in Table 1. Dates of visual rating of HLB severity and canopy health are shown in Table 2.

### 2.3. UAS Data Acquisition and Image Processing

#### 2.3.1. UAS Data Acquisition

Dates of UAS data collection are shown in Table 2. We flew a DJI Phantom 4 Pro (SZ DJI Technology Co., Ltd., Shenzhen, China) equipped with a 20-megapixel, one-inch complementary metal oxide semiconductor (CMOS) red, green, and blue (RGB) camera to capture a visible spectrum. In addition, a DJI Phantom 4 multispectral (SZ DJI Technology Co., Ltd., Shenzhen, China), attached with a Multispectral sensor, which includes five bands (blue (450 ± 16 nm) green (560 ± 16 nm), red (650 ± 16 nm), red-edge (730 ± 16 nm), and near-infrared (740 ± 26 nm)), was used. This platform has an integrated spectral sunlight sensor that captures solar irradiance on top of the drone. The flight altitude is crucial when it comes to collecting UAS imagery. According to the literature review, varying flight altitudes have been used to detect HLB in citrus, including 50 [13], 60 [16], and 100 m [15]. In our study, a higher spatial resolution was required as we were trying to assess the five different levels of HLB severity and canopy health. Therefore, the platforms were flown at 40 m above ground level, which was lower altitude than the previous articles, with front and side overlap settings of 80% to ensure high spatial resolution. In addition, the two different flight altitudes were performed to assess the effect of varying spatial resolution on VIs and TFs. The flight mission was performed when the wind speed was less than 10 miles per hour and on a sunny day between 10 AM and 2 PM to ensure the sun elevation angle was the highest during the data collection. Also, images needed for radiometric calibration were collected before and after the multispectral UAS survey using a radiometric calibration panel. Furthermore, the ground control points (GCPs) were marked with checkered boards, each assigned a distinct identification number and placed at each corner of the study area before conducting the UAS survey. The V-map GNSS receiver (Micro Aerial Projects, Gainesville, FL, USA) was then used to obtain the accurate coordinates of each GCP, which was further used for georeferencing.

#### 2.3.2. Image Processing

Orthomosaic generation

The individual UAS data were used to generate orthomosaics through the Structure from motion (SfM) process for 3D reconstruction of the set of overlapping UAS images. We used Agisoft Metashape software version 1.7.1 (Agisoft LLC, St. Petersburg, Russia) to process the UAS-RGB images captured by the DJI Phantom 4 Pro platform. The pipeline reported by Bhandari et al. [25] was followed. The multispectral images acquired by the DJI Phantom 4 multispectral platform were processed using DJI Terra software version 2.3.0 (SZ DJI Technology Co., Ltd., Shenzhen, China). When processing with the DJI Terra software, radiometric calibration was automatically performed [26]. The coordinates of each GCP were incorporated during generating orthomosaic, aiming to improve geometrical accuracy among multi-temporal UAS flights.

Vegetation indices (VIs) extraction

QGIS software version 3.34.10. was used to extract VIs, and 14 VIs were used in this study (Table 3). These VIs were chosen as they have been found to be promising in assessing HLB disease such as normalized difference red-edge index (NDRE), normalized difference vegetation index (NDVI), and red-edge chlorophyll index (CI) [13], while most VIs were found to be highly correlated with chlorophyll [27], which is associated with chlorosis, a common symptom of HLB. In addition, canopy cover (CC) was also extracted using the Canopeo algorithm introduced by Patrignani and Ochsner [28].

Textural features (TFs) extraction

TFs calculation and extraction were also performed using QGIS software version 3.34.10. The “r.recode” algorithm in QGIS was used to re-classify digital numbers (DNs) into 20 classes for each band (G, B, R, RE, and NIR). The reclassified raster of each band was further used for TFs extraction using the “r.texture” function. The “r.texture” uses the common texture model based on the grey-level co-occurrence matrix as described by Haralick [42]. The default parameter of moving window size of 3 and a distance of 1 between two samples was used. Five texture measurement methods, including angular second moment (ASM), contrast (Contr), correlation (Corr), variance (Var), and inverse difference moment (IDM), were used. The five texture measurement methods combined with the five spectral bands resulted in 25 TFs variables.

### 2.4. Data Analysis

#### 2.4.1. Analysis of Variance (ANOVA)

An analysis of variance according to RCBD (3 treatments and 3 blocks) design was carried out to determine the effect of the injected antimicrobial on UAS-based VIs of the HLB-affected citrus trees. Additionally, one-way ANOVA was used to test if there were significant differences in UAS-based features among several categories of visual rating. We used the computed F-value to identify the best feature in differentiating HLB severity and canopy parameters, as the higher computed F-value is, the stronger the significance (lower *p*-value) is.

#### 2.4.2. Classification of HLB Severity Using Machine Learning

As there were limited samples for the 1 and 4 scales of HLB severity (Figure 2a), we recategorized them from four to two categories by merging the 1 and 2 scales as mild HLB symptoms and the 3 and 4 scales as severe HLB symptoms (Figure 2a). So, the classification task in this study was to assess the performance of models in accurately classifying the mild and severe HLB-affected citrus. RStudio version 2024.04.2 was used to perform these classification tasks. Pearson’s correlogram was graphically analyzed to assess the association among predictor variables using the package “corrplot”. For classification, there was an imbalanced dataset between the two HLB severity levels (Figure 2a). When it comes to dealing with imbalanced datasets, using an oversampling strategy can improve the performance of a machine-based model [43]. Therefore, to handle this challenge, we used the “ROSE” package to oversample during the model being trained, aiming to minimize a bias of the generated models. Overall, 80% of the entire dataset (482 data points) was assigned to a training dataset. Four machine learning algorithms, including naive bayes (NB), random forest (RF), support vector machine (SVM), and eXtreme gradient boosting (XGBoost), were adopted to establish the classification models. The following packages, including “naivebayes”, “randomForest”, “e1071”, and “xgboost”, were used to build NB, RF, SVM, and XGBoost, respectively. In addition, feature selection was performed using the “Boruta” package. The validation of the built models was carried out using 20% unseen data (104 data points). Confusion matrix was created using the package “caret” in RStudio. Accuracy matrices, as shown in Figure 2c, were calculated to assess the accuracy of each model. The pipeline of this study is presented in Figure 3.

## 3. Results

### 3.1. Assessing the Effect of the Injected OTC on the HLB-Affected Citrus Using UAS-Based Remote Sensing

The HLB-affected citrus trees injected with OTC did not show significant differences (*p* > 0.05) for almost all UAS-based VIs in nearly all dates of data collection except for CC collected in June (Table 4). However, this significant difference seemed to be due to citrus phenology rather than the effects of the OTC injection, as only 2% differences were found. Moreover, before OTC was applied, 1.40% and 1.23% of the CC of the citrus trees injected with control treatments (water and mock, respectively) were higher than the ones injected with OTC (Table 5). Despite only a 2% difference, a significant difference was found because there was a significantly lower variation among replications, resulting in the lowest CV (0.57%), while variation among treatments remained the same. Overall, UAS-based VIs showed no significant differences among the treatments applied. The *p*-value is used to determine if the effect of treatment administered to the citrus trees is significantly different from one to another, so the trend of such values can also provide information regarding the effect of treatments. The results show that the *p*-value has fluctuated across dates of data collection, and a downtrend of the P-value was not detected after up to 14 months of the injection (from February 2023 to April 2024), indicating the antimicrobials injected into the HLB-affected citrus trees did not have any effect on UAS-based VIs.

### 3.2. Assessing HLB Severity and Canopy Health of Citrus via UAS-Based Remote Sensing

The UAS-based remote sensing features were used to differentiate the severity levels of HLB disease and canopy health observed by the specialists. Overall, VIs outperformed TFs in distinguishing the levels of HLB severity and canopy health of the HLB-affected citrus trees (Figure 4). No TFs were significantly different (*p* > 0.01) among the four categories of HLB severity (Figure 4a), indicating that the four HLB severity levels had relatively the same TFs. In contrast, there were significant differences (*p* < 0.01) of 10 VIs, including CC, CI, ExG, GCI, GNDVI, IPVI, LCI, NDRE, NDVI, and NGRDI, among the four categories, indicating that those VIs were statistically able to differentiate the levels of HLB severity. Based on the computed F-value, the CI index was the best indicator for distinguishing HLB severity (Figure 4a). A mildly HLB-affected citrus tree tended to have a higher CI index than a severely HLB-affected one and vice versa (Figure 5a). Moreover, both VIs and TFs were statistically able to distinguish canopy color and canopy density. The performance of VIs outperformed that of TFs for both canopy color and canopy density. Like HLB severity, the CI index appeared to be the best in assessing canopy color, followed by NDRE and LCI index, indicating that the greener canopy had a greater CI index value and vice versa (Figure 4b and Figure 5b). Likewise, the CC was outstanding in assessing the different levels of canopy density. The CC of very sparse citrus tree was only 81.2%, compared to 97.9% of very dense canopy (Figure 4c and Figure 5c).

Figure 6 shows individual trees and the visualization of some VIs and TFs of the HLB-affected citrus with 0% compared to 50–75% of citrus tree branches with HLB symptoms. The differences between mild and severe HLB-affected citrus trees could be clearly visualized using the outstanding HLB disease indices (CI, NDRE, and CC index). Based on the results of this study, the following indices, the NDRE, CI, and CC index, were considered promising features for monitoring citrus canopy health (Figure 7). The result shows two months (February and August 2023) where the overall values of the UAS-based VIs were lower than the other months for all three VIs. The highest NDRE and CI values were observed in April 2023. Additionally, CC derived on April 2024 was the highest value and appeared to be the most consistent CC among all flight missions.

### 3.3. Classification of Huanglongbing Severity Using UAS-Based Remote Sensing

Pearson’s correlation among predictor variables is presented in Figure 8. Overall, a higher association among themselves was observed in TFs than in VIs. As expected, a lower association or even uncorrelation between TFs and VIs was observed compared to those among themselves. This result indicates that a combination of TFs and VIs could bring new model variations.

Overall, the VIs-based models slightly overperformed the TFs-based models for HLB severity classification, except for specificity (Figure 9). UP to 0.93 accuracy was observed for binary classification of HLB severity. Additionally, up to 0.97 and 0.98 were observed for recall and precision, respectively (Table 6). However, the specificity and negative predictive value, designed to assess the accuracy of severe HLB detection, were poor, ranging from 0.34 to 0.84 and from 0.14 to 0.65, respectively. Moreover, the results show that the fusion of TFs and VIs models could improve all accuracy metrics (accuracy, recall, precision, specificity, and negative predictive value) for all models (Table 6 and Figure 9). The performance of RF, SVM, and XGBoost was quite similar to each other. Also, their accuracy was impressive in accurately detecting true mild HLB severity (>0.9), in addition to maintaining an acceptable accuracy when encountering severe HLB ones. In the other case, the NB model yielded several false severe HLB points, resulting in inferior accuracy, recall, and especially negative predictive value, although its specificity was the highest (Figure 10 and Table 6).

### 3.4. Effect of Varying Flight Altitudes on Vegetation Indices (VIs) and Textural Features (TFs)

The flight parameters and required resources for both flight altitudes are shown in Table 7. The flight altitude of 30 m with a ground sampling distance (GSD) of 1.5 cm/pixel required a flight duration, image-processing time, and storage size roughly twice of those of the flight altitude of 40 m with GSD of 2.0 cm/pixel. The VIs, including the CI, NDRE, and NDVI index, showed a strong relationship when they were captured at different flight altitudes (30 and 40 m) with coefficients of determination (R^2^) of 0.91, 0.91, and 0.95, respectively. On the other hand, TFs were heavily affected by spatial resolution as low R^2^ values were observed for all three TFs (Figure 11).

## 4. Discussion

For growers, yield is the most critical and relevant indicator of assessing HLB severity. Still, it could be measured only at a certain time in addition to its labor-intensive nature [7]. Moreover, a reliable assessment of the effect of HLB therapies is essential. Recently, Levy et al. [7] reported that canopy health is more relevant for yield than the CLas titers in assessing the impacts of therapies in Florida. Furthermore, canopy density could be used to evaluate HLB severity. In this study, we used several UAS-based VIs to assess the effectiveness of the OTC injected into the HLB-affected citrus trees in Texas. No significant differences in UAS-based features were observed in our trials in Texas. A manual visual rating also provided the same conclusion. These are possible explanations. The HLB disease pressure and rate of spread in Texas are much lower than in Florida due to varying environmental conditions, varieties, and production practices [10]. The HLB-affected citrus trees in Texas are also relatively healthy and more productive than their Florida counterparts. Hence, the propensity of OTC to induce large and significant differences in the measured tree health traits may be minimal. However, UAS-based VIs agreed with the visual rating of HLB severity, canopy color, and canopy density observed by the specialists, where CI and CC index were statistically able to differentiate the severity levels and canopy health of the HLB-affected citrus trees. This highlights the potential of using the proposed UAS-based assessment to assess the severity of HLB disease. Moreover, our results confirmed the previous findings by Chang et al. [13], who found that the CI and NDRE were outstanding indices in differentiating HLB-positives and HLB-negatives of citrus trees. Moreover, Chang et al. [13]. and Garcia-Ruiz et al. [15] also reported that red edge-related VIs showed more capability for citrus disease monitoring, comparable to our findings.

Despite its subjectivity, the field assessment of HLB-affected severity is typically conducted, but to obtain reliable data, it needs to be performed by experienced specialists. However, many farmers do not have access to specialists [44]. Therefore, a trained classification model that leverages specialist-based HLB severity and UAS-based remote sensing features could be one of the solutions. This model is designed to predict unseen events from what has been learned during training [45]. So far, however, we are unaware of any reporting of using UAS-based remote sensing features to classify HLB severity. All previous reports have focused on differentiating HLB-affected and HLB-free citrus. For example, accuracy ranging from 0.67 to 0.85 was reported by Garcia-Ruiz et al. [15], and accuracy ranging from 0.72 to 1.00 was reported by Lan et al. [16]. Their HLB-positive detection accuracy was comparable to our HLB severity classification, ranging from 0.58 to 0.93. The innovation in the article reported by Lan et al. [16] is that they used a threshold strategy, together with machine learning algorithms, where their proposed approach achieved up to 1.00 accuracy. Our research, for the first time, demonstrated the potential of UAS-based remote sensing utilizing both VIs and TFs, alongside with machine learning, to accurately classify the severity levels of HLB, where we obtained accuracy up to 0.93 when VIs and TFs were combined for model establishment. All classification models yielded lower accuracy when the HLB-severe citrus was classified as specificity, and the negative predictive value of all models were poor. This suggests that the models struggle to accurately recognize the HLB-severe citrus. The dataset was extremely imbalanced among the HLB severity levels (Figure 2a), with more mild symptom data points than severe ones. When models encounter this challenge, they are likely to more accurately detect mild HLB than severe HLB, resulting in a higher recall and precision than specificity and negative predictive value. Our findings were similar to the previous findings, where it was found that the fusion of VIs and TFs significantly improved the accuracy of the predictive model [23,46,47]. Moreover, this result aligned well with those reported by Terensan et al. [17], who found that the classification of the severity of bacterial leaf blight in paddy fields could be improved by combining spectral and textural data. A scientific explanation for the improved accuracy is the fact that TFs take the spatial variation in the pixels into account, resulting in providing additional information about the physical structure of the canopy, edges of a canopy, and overall canopy architecture [23].

Increasing the UAS flight altitude resulted in a shorter time for image acquisition and processing in addition to lower storage requirement, but the trade-off for this was a lower spatial resolution. Our result agreed with those reported by Njane et al. [48], who found an R^2^ of 0.99 for NDVI captured at 15 and 30 m. This indicates that the spectral information is not significantly sensitive to a negligible difference of spatial resolution, so a higher flight altitude would be recommended, especially when capturing a large area, as it saves time and resources. Moreover, a shorter flight duration reduces the possibility of uneven weather conditions during flight missions, such as wind speed and cloud movement. On the contrary, this robust relationship was not observed by TFs when capturing at different flight altitudes. This was not surprising because the textural information is a spatial feature [49], and its result depends heavily on the spatial relationship [50]. Textural information of rice aboveground biomass [51] and wheat growth parameters [52] strongly depended on the spatial resolution, which was comparable to our results.

One of the essential limitations of the present study was using only two severity levels, e.g., mild and severe ones. Another limitation was that we trained our machine learning models using the severity levels of HLB based on visual rating, which is subjective. For future work, the classification of multi HLB severity levels should be investigated. Additionally, it is interesting if a higher spatial resolution could improve the efficacy of the proposed approach in assessing HLB severity and canopy health. Furthermore, the development of models using other assessments that are less subjective, such as quantifying CLas with PCR, microscopic or spectroscopic, is also interesting. More importantly, transfer learning techniques that significantly leverage model generalizability need to be adopted, especially when classifying HLB severity levels across cultivars, years, and locations.

## 5. Conclusions

In this study, we used UAS-based VIs and TFs to assess the effectiveness of oxytetracycline (OTC) on HLB-affected citrus trees and to differentiate between varying levels of HLB severity and canopy health. UAS images were collected over the experimental field in Weslaco, Texas, along with treatment information and field identification of the HLB-affected citrus trees by experts. The UAS data were pre-processed, and orthomosaic images were generated. Individual trees were identified in these images, and VIs and TFs were extracted. These features were input variables for the four machine-learning algorithms to develop the classification model. Based on the results obtained in this study, the effect of injected OTC on UAS-based VIs of the HLB-affected citrus trees in Texas was minimal. However, the CI index was able to differentiate the visual-based HLB severity levels and canopy color. Additionally, the CC appeared to be an outstanding index in distinguishing the canopy density of citrus. These VIs would be recommended for monitoring HLB severity in citrus. A combination of VIs and TFs improved classification accuracy for all accuracy metrics and models for HLB severity classification. In addition, RF and XGBoost models were promising in classifying HLB severity levels. Our results highlight the potential of using UAS-based features in assessing the severity of HLB-affected citrus trees.

## Figures and Tables

**Figure 1 sensors-24-07646-f001:**
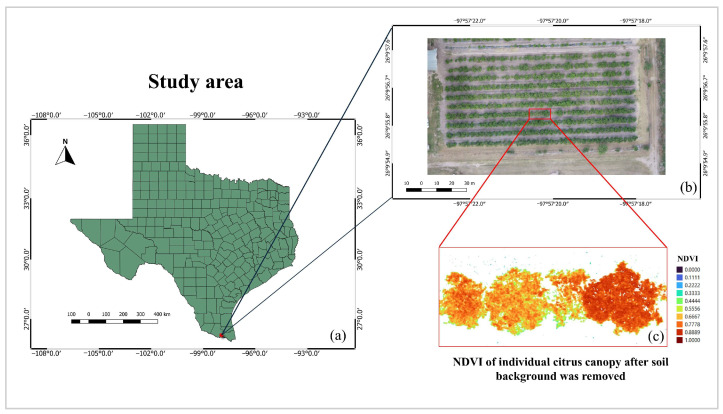
Study area in this study, located at Weslaco, Texas: (**a**) the map of Texas showing all the counties and Hidalgo County highlighted in red, (**b**) a sample of the citrus trees studied in red rectangle, and (**c**) NDVI of the selected sample of citrus trees showing their canopy after the soil background was removed.

**Figure 2 sensors-24-07646-f002:**
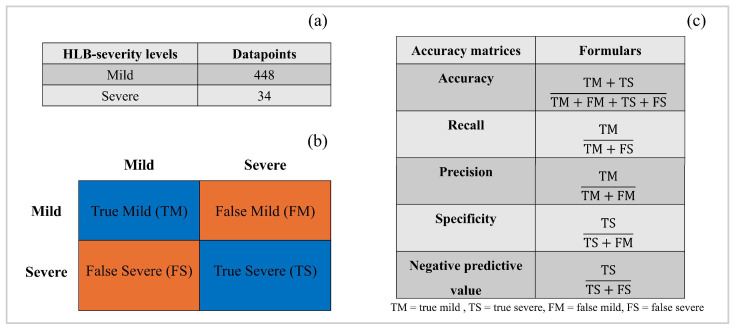
Number of data points of huanglongbing (HLB) severity levels (**a**), confusion matrix (**b**), and accuracy matrices (**c**) used to assess accuracy in this study.

**Figure 3 sensors-24-07646-f003:**
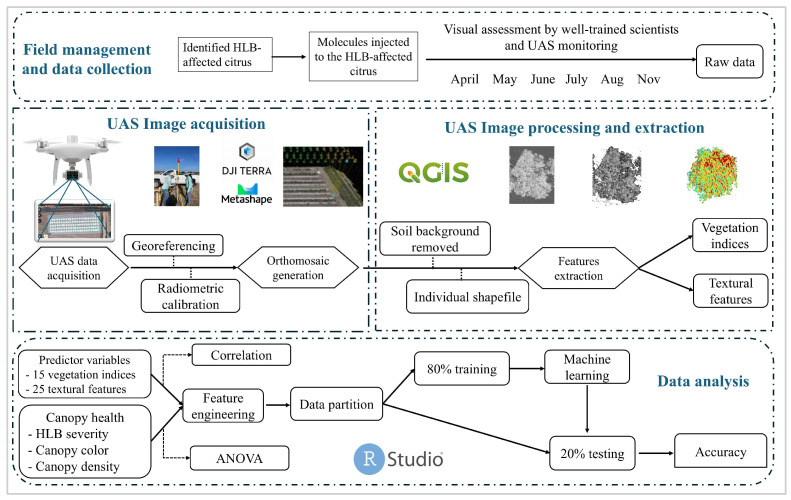
The flowchart of unmanned aerial system (UAS)-based remote sensing in assessing huanglongbing (HLB) severity and canopy health.

**Figure 4 sensors-24-07646-f004:**
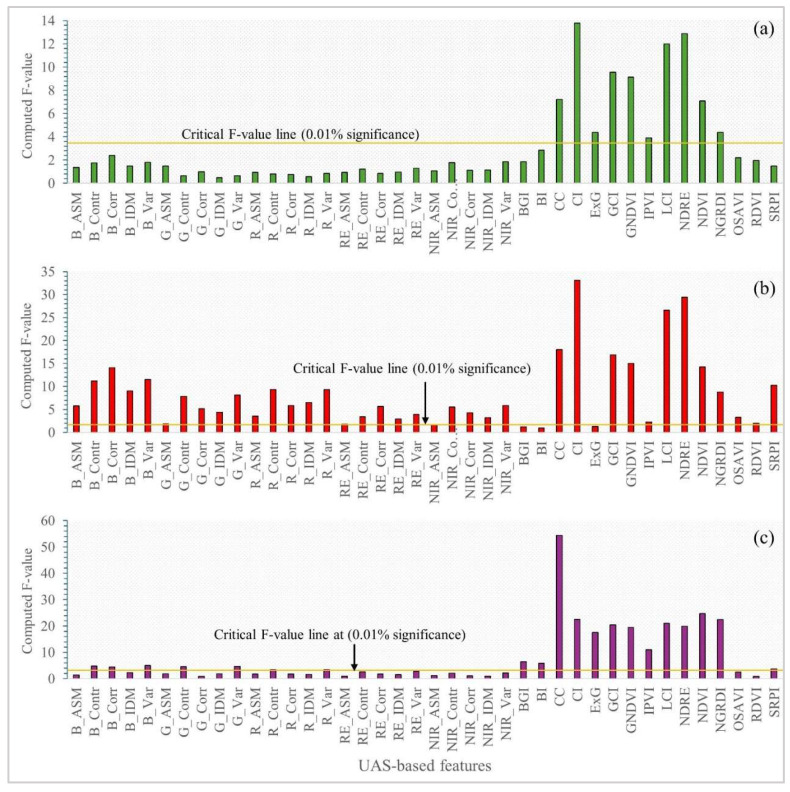
Computed F-values of unmanned aerial system (UAS)-based features among different categories of huanglongbing severity (**a**), canopy color (**b**), and canopy density (**c**).

**Figure 5 sensors-24-07646-f005:**
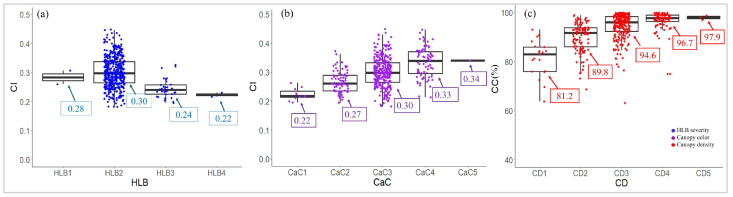
Distribution of vegetation indices among visual rating categories for huanglongbing (HLB) severity (**a**), canopy color (**b**), and canopy density (**c**). The values in the boxes show the mean values of that category (*n* = 482). The subfigure in the (**c**) is a dot legend.

**Figure 6 sensors-24-07646-f006:**
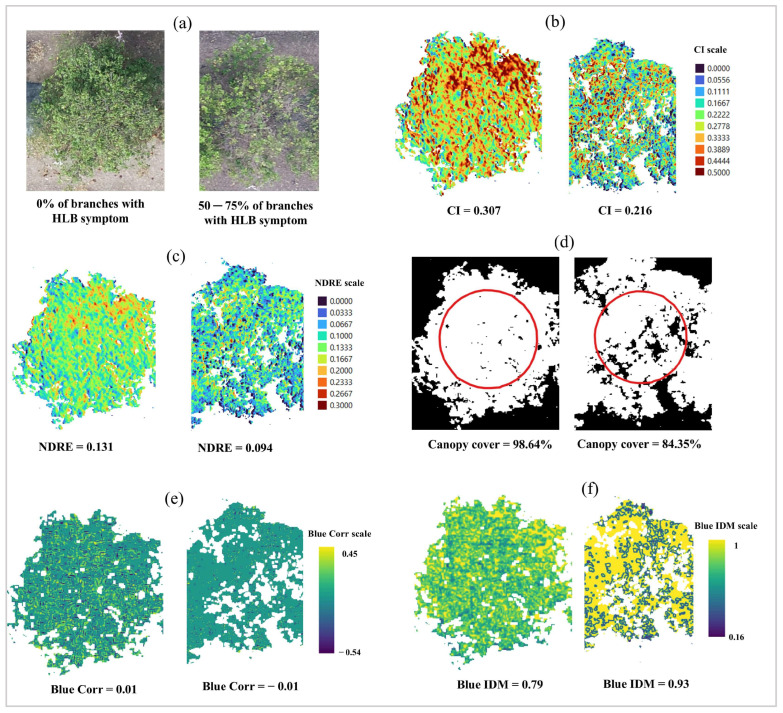
Aerial image captured by an unmanned aerial system (UAS) of a mild huanglongbing (HLB) and severe one (**a**), and their vegetation indices (**b**–**d**) and textural features (**e**,**f**). Spatial resolution is 2 cm. The red circles in the (**d**) are shapefile used to extract canopy cover.

**Figure 7 sensors-24-07646-f007:**
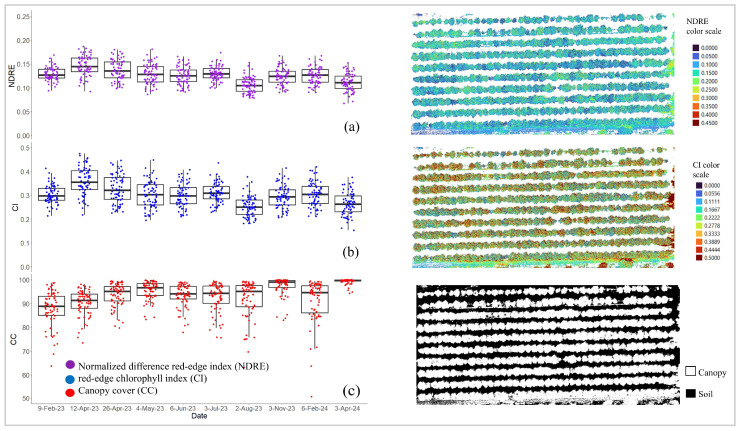
Boxplots of normalized difference red-edge index (**a**), red-edge chlorophyll index (**b**), and canopy cover captured throughout the growing season and their vegetation indices (VIs) maps captured on 3 April 2024. Spatial resolution is 2 cm. The subfigure in the (**c**) is a dot legend.

**Figure 8 sensors-24-07646-f008:**
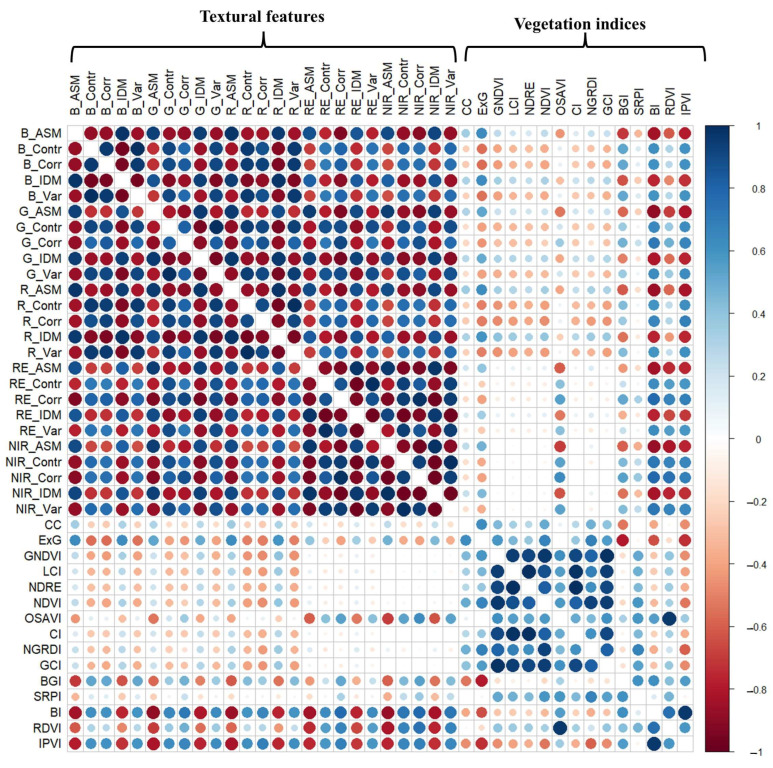
Pearson’s correlation among predictor variables (*n* = 482).

**Figure 9 sensors-24-07646-f009:**
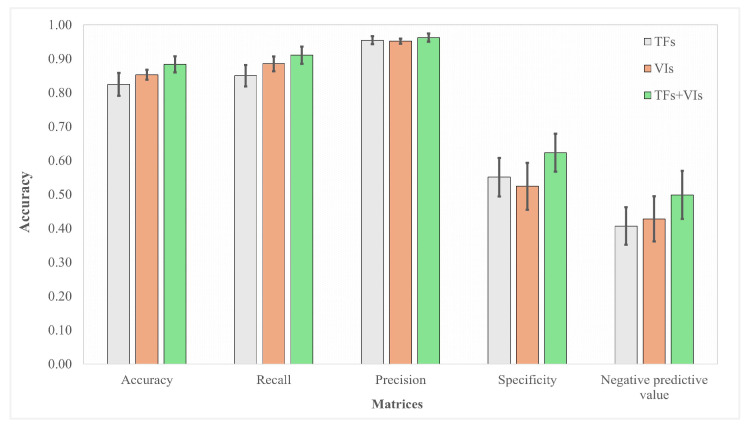
Comparison of accuracy matrices among different predictor variables.

**Figure 10 sensors-24-07646-f010:**
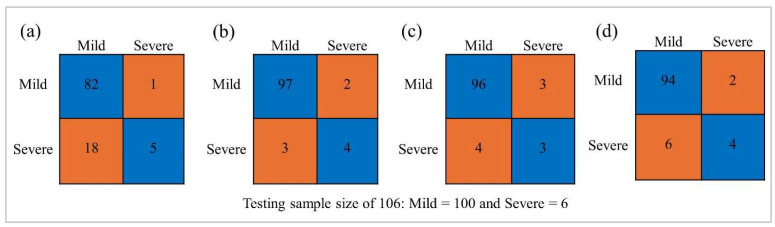
Confusion matrix of naive bayes (**a**), random forest (**b**), support vector machine (**c**), and eXtreme gradient boosting (**d**) derived from the fusion of textual features and vegetation indices model.

**Figure 11 sensors-24-07646-f011:**
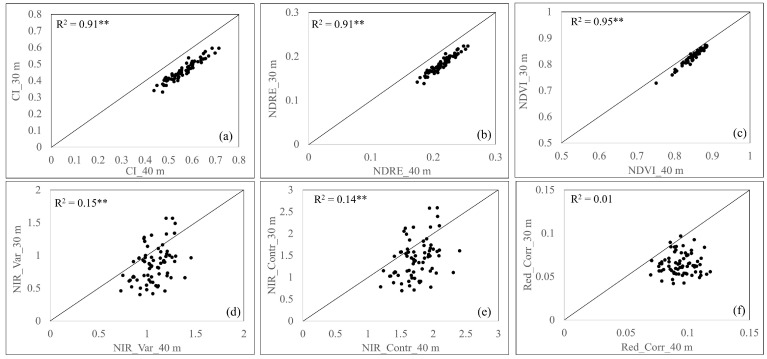
Comparison between different flight altitudes (30 and 40 m) for vegetation indices including CI (**a**), NDRE (**b**), and NDVI (**c**) and textural features including NIR variance (**d**), NIR contrast (**e**), and red correlation (**f**) (n = 72). ** indicates significant differences at 0.01%.

**Table 1 sensors-24-07646-t001:** Visual rating assessment used to evaluate huanglongbing (HLB) severity and canopy health of the HLB-affected citrus in this study.

Scale	Description		
	HLB Severity	Canopy Color	Canopy Density
1	0% of branches with HLB symptom	very yellow	very sparse
2	0–25% of branches with HLB symptom	yellow	sparse
3	25–50% of branches with HLB symptom	green	good balance between leaf volume and canopy aeration
4	50–75% of branches with HLB symptom	very green	density
5	>75% of branches with HLB symptom	dark green	very density

**Table 2 sensors-24-07646-t002:** The date of unmanned aerial system (UAS) data collection and visual rating of huanglongbing (HLB) severity and canopy health of the HLB-affected citrus.

Flight No.	UAS Flights	Visual Rating
1	9 February 2023	9 February 2023
2	12 April 2023	-
3	26 April 2023	27 April 2023
4	4 May 2023	10 May 2023
5	6 June 2023	10 June 2023
6	3 July 2023	10 July 2023
7	2 August 2023	10 August 2023
8	3 November 2023	11 November 2023
9	6 February 2024	-
10	3 April 2024	-

**Table 3 sensors-24-07646-t003:** Vegetation indices (VIs) used in this study.

Vegetation Indices	Formula ^†^	Reference
Excess green index	ExG = (2 × G) − R − B	[29]
Green normalized difference vegetation index	GNDVI = (NIR − G)/(NIR + G)	[30]
Leaf chlorophyll index	LCI = (NIR − RE)/(NIR + R)	[31]
Normalized difference red-edge index	NDRE = (NIR − RE)/(NIR + RE)	[32]
Normalized difference vegetation index	NDVI = (NIR − R)/(NIR + R)	[33]
Optimized soil-adjusted vegetation index	OSAVI = (1 + 0.16) × (NIR − R)/(NIR + R + 0.16)	[34]
Red-edge chlorophyll index	CI = (NIR/RedEdge) − 1	[35]
Normalized green red difference index	NGRDI = (G − R)/(G + R)	[36]
Green chlorophyll index	GCI = (NIR/Green) − 1	[35]
Blue green pigment index	BGI = B/G	[37]
Simple ratio pigment index	SRPI = B/R	[38]
Brightness index	BI = √((R^2^ + G^2^ + B^2^)/3)	[39]
Renormalized difference vegetation index	RDVI = NIR ─ R/√NIR + R	[40]
Infrared percentage vegetation index	IPVI = NIR/NIR + R	[41]

^†^ B, G, R, RE, and NIR represent the blue, green, red, red-edge, and NIR bands, respectively, in the above formulas.

**Table 4 sensors-24-07646-t004:** *p*-value for the effect of the injected antimicrobials on UAS-based VIs of the HLB-affected citrus. The first column represents data before treatment, while the subsequent columns represent data after treatment.

Vegetation Indices	Dates of Data Collection ^†^									
	February (Before Treatment)	12 April 2023	26 April 2023	4 May 2023	6 June 2023	3 July 2023	2 August 2023	3 November 2023	6 February 2024	3 April 2024
CC	0.52	0.54	0.22	0.17	0.01	0.29	0.51	0.06	0.75	0.38
ExG	0.66	0.11	0.08	0.09	0.07	0.42	0.51	0.35	0.55	0.71
GNDVI	0.29	0.83	0.83	0.79	0.37	0.59	0.33	0.50	0.25	0.38
LCI	0.31	0.75	0.74	0.63	0.38	0.53	0.12	0.22	0.10	0.75
NDRE	0.32	0.75	0.73	0.60	0.35	0.51	0.06	0.25	0.09	0.29
NDVI	0.28	0.73	0.76	0.75	0.66	0.55	0.30	0.26	0.30	0.32
OSAVI	0.52	0.86	0.92	0.71	0.11	0.69	0.50	0.74	0.66	0.43
CI	0.32	0.73	0.73	0.59	0.36	0.50	0.06	0.30	0.08	0.34

^†^ *p*-value > 0.05 indicates nonsignificant and *p*-value < 0.05 indicates significant at 0.05%.

**Table 5 sensors-24-07646-t005:** Mean ± standard deviation of canopy cover of the HLB-affected citrus before and after treatment with different antimicrobials. The first column represents data before treatment, while the subsequent columns represent data after treatment.

Treatment	Dates of Data Collection									
	February (Before Treatment)	12 April 2023	26 April 2023	4 May 2023	6 June 2023 ^†^	3 July 2023	2 August 2023	3 November 2023	6 February 2024	3 April 2024
Arbor-OTC	86.4 ± 4.7	90.1 ± 3.4	92.9 ± 3.3	94.6 ± 2.7	92.3 ± 1.3 ^b^	91.2 ± 1.1	90.7 ± 3.7	97.7 ± 0.8	91.4 ± 6.7	99.4 ± 0.3
water	87.8 ± 2.2	91.0 ± 2.8	94.0 ± 2.7	95.8 ± 1.8	94.2 ± 1.8 ^a^	93.7 ± 2.3	92.6 ± 3.5	98.2 ± 1.0	91.6 ± 1.7	99.5 ± 0.3
DMSO	87.63 ± 3.0	90.6 ± 2.2	94.2 ± 2.5	96.2 ± 1.1	94.7 ± 1.8 ^a^	93.8 ± 2.0	94.7 ± 2.2	99.3 ± 0.2	94.0 ± 1.0	99.7 ± 0.2
CV (%)	1.69	1.00	0.86	0.89	0.57	2.12	4.23	0.59	4.95	0.26

^†^ Means in the same column with the same letters are not significantly different by LSD at *p* ≤ 0.05.

**Table 6 sensors-24-07646-t006:** Accuracy of classification models derived from UAS-based textural features (TFs) and vegetation indices (VIs) with different machine learning algorithms.

Predictor Variables	Models ^†^	Accuracy	Recall	Precision	Specificity	Negative Predictive Value
TFs	NB	0.58 ± 0.08	0.54 ± 0.08	0.97 ± 0.03	0.84 ± 0.13	0.14 ± 0.02
	RF	0.90 ± 0.02	0.95 ± 0.01	0.94 ± 0.01	0.34 ± 0.05	0.42 ± 0.11
	SVM	0.92 ± 0.03	0.95 ± 0.03	0.96 ± 0.01	0.58 ± 0.11	0.54 ± 0.16
	XGBoost	0.91 ± 0.03	0.96 ± 0.02	0.95 ± 0.02	0.46 ± 0.14	0.52 ± 0.11
VIs	NB	0.69 ± 0.05	0.68 ± 0.06	0.97 ± 0.01	0.81 ± 0.06	0.19 ± 0.05
	RF	0.92 ± 0.01	0.97 ± 0.02	0.95 ± 0.02	0.40 ± 0.14	0.58 ± 0.19
	SVM	0.90 ± 0.02	0.94 ± 0.03	0.95 ± 0.01	0.46 ± 0.12	0.42 ± 0.12
	XGBoost	0.89 ± 0.02	0.95 ± 0.02	0.93 ± 0.02	0.38 ± 0.12	0.47 ± 0.14
VIs + TFs	NB	0.76 ± 0.07	0.76 ± 0.08	0.98 ± 0.02	0.80 ± 0.10	0.23 ± 0.05
	RF	0.93 ± 0.02	0.96 ± 0.03	0.96 ± 0.02	0.58 ± 0.13	0.65 ± 0.22
	SVM	0.92 ± 0.00	0.96 ± 0.02	0.96 ± 0.02	0.55 ± 0.14	0.52 ± 0.09
	XGBoost	0.92 ± 0.04	0.96 ± 0.01	0.95 ± 0.04	0.55 ± 0.22	0.59 ± 0.07

^†^ NB = naive bayes, RF = random forest, SVM = support vector machine, XGBoost = eXtreme gradient boosting.

**Table 7 sensors-24-07646-t007:** Comparison of flight parameters, required time, and storage capacity for both flight altitudes for a single flight (study area of 1.23 ha).

Flight Altitudes	Spatial Resolution	Flight Time	Number of Images	Raw Image Storge Size	Generating Orthomosaic Time ^†^
30 m	1.5 cm	20 min	2900	9.81 GB	64 min
40 m	2.0 cm	10 min	1618	5.47 GB	36 min

^†^ Processed by DJI Terra software using a desktop with the following specifications: CPU: Intel^®^ Core™ i9-9900K CPU @3.60GHz (Intel, Santa Clara, CA, USA), RAM: 64 GB, and GPU: NVIDIA GeForce GTX 1660 SUPER (NVIDIA, Santa Clara, CA, USA).

## Data Availability

Data are contained within the article.

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
