# Peer review of "Assessing Huanglongbing Severity and Canopy Parameters of the Huanglongbing-Affected Citrus in Texas Using Unmanned Aerial System-Based Remote Sensing and Machine Learning"

_sensors, 2024, doi:10.3390/s24237646_

Round 1
Reviewer 1 Report
Comments and Suggestions for Authors
1.From your article, I learned that huanglongbing is a great harm to citrus, but why should we distinguish its severity? And do you have any basis for the division you mentioned in Table1? I suggest you explain it more clearly.
2.Please specify the data collected for each period in Table 2. I think you are not clear description because 14 months mentioned in Line 234.This kind of messy data is2 a little difficult to realize
3.This article achieve the good results, the highest accuracy reached about 0.92. But you also mentioned in line 352 that "and accuracy ranging from 0.72 to 1.00 was reported by Lan et al." So what's the innovation in this article?
Author Response
1. From your article, I learned that huanglongbing is a great harm to citrus, but why should we distinguish its severity? And do you have any basis for the division you mentioned in Table1? I suggest you explain it more clearly. |
Thank you for pointing that out, and we agree with your comments. Therefore, more detail on the importance of HLB severity assessment was clearly explained in line 62 – 68. In addition, we followed the criteria reported by Archer et al. (2022) to perform visual rating of HLB for the division of Table 1. (line 113 – 116) |
2.Please specify the data collected for each period in Table 2. I think you are not clear description because 14 months mentioned in Line 234. This kind of messy data is2 a little difficult to realize |
Thank you for pointing that out and here is the clarification. After the treatments were injected into HLB-affected citrus, we observed visual ratings and flew the UAS over the trial targeting by monthly. For various reasons, we were not able to collect the data monthly as planned (line 113-114). The 14 months (mentioned in line 240) were counted from the treatments injection (February 2023) to the last month of UAS data collection (April 2024). line 240. |
3.This article achieve the good results, the highest accuracy reached about 0.92. But you also mentioned in line 352 that "and accuracy ranging from 0.72 to 1.00 was reported by Lan et al." So what's the innovation in this article? |
Our work, for the first time, reported using UAS-based remote sensing approach to assess the severity of HLB. As has not been reported before, we could not find the same work to make such a comparison of accuracy. Therefore, we compared our results with the most similar one instead. Research, done by Lan et al., was very similar to ours where they used UAS-based remote sensing to detect HLB positive, while we were using UAS-based remote sensing to assess HLB severity (line 357 – 361). An innovation in their work was that their approach (AdaBoost) achieved up to 100% accuracy for HLB detection. |

Reviewer 2 Report
Comments and Suggestions for Authors
The paper proposes to obtain visible images and multispectral images by UAS, and then extract VI and TF indexes from these images to evaluate HLB and its impact on citrus. Although the method used in the paper is not particularly novel, the paper describes the entire research process in detail and can provide a useful reference for relevant researchers. The author mentioned in the paper "Our results highlight the potential of using UAS-based features in assessing the severity of HLB-affected citrus trees", and I agree with the author's evaluation of this achievement.
Since the author used the TF index, the spatial resolution of the images obtained by the UAS and the visualization results of the TF index extracted from the images should be given in the paper, but no information about the TF index is given in Figures 6 and 7.
Since the focus of this study is the remote sensing method, and the VI and TF indicators are closely related to the flight altitude of the UAS, it is recommended that the author supplement the results of data processing based on different flight altitudes, so as to provide the best data collection plan.
Author Response
1. The paper proposes to obtain visible images and multispectral images by UAS, and then extract VI and TF indexes from these images to evaluate HLB and its impact on citrus. Although the method used in the paper is not particularly novel, the paper describes the entire research process in detail and can provide a useful reference for relevant researchers. The author mentioned in the paper "Our results highlight the potential of using UAS-based features in assessing the severity of HLB-affected citrus trees", and I agree with the author's evaluation of this achievement. |
Thank you for your comments |
2. Since the author used the TF index, the spatial resolution of the images obtained by the UAS and the visualization results of the TF index extracted from the images should be given in the paper, but no information about the TF index is given in Figures 6 and 7. |
We agree with your comments, so a spatial resolution of 2 cm has been already added in the revised manuscript (line 280 and 294) |
3. Since the focus of this study is the remote sensing method, and the VI and TF indicators are closely related to the flight altitude of the UAS, it is recommended that the author supplement the results of data processing based on different flight altitudes, so as to provide the best data collection plan. |
It is a very interesting topic if a higher resolution could improve accuracy and efficacy of this approach. However, that was not the objective of this manuscript. Therefore, we have included this recommendation as future research in the revised manuscript (line 381 – 382)
Nevertheless, the logical explanation of why we have chosen of flying at 40 m above ground levels was given in line 131 – 137. |
Round 2
Reviewer 1 Report
Comments and Suggestions for Authors
I have no more questions.
Author Response
Reviewer 1 has no more questions. Therefore, we did not respond.
Reviewer 2 Report
Comments and Suggestions for Authors
The author has only made brief revisions to the previous review comments and has not added any additional experiments. In this case, I still maintain my last review opinion.
Author Response
Since the author used the TF index, the spatial resolution of the images obtained by the UAS and the visualization results of the TF index extracted from the images should be given in the paper, but no information about the TF index is presented in Figures 6 and 7.
Spatial resolutions have been provided in lines 285 and 295. Some TFs visualizations have been added in Figure 6 (279-283). However, based on the results from Figure 4 (lines 254-258), TFs underperformed VIs in assessing HLB severity, so they are not recommended for monitoring. Hence, we omit TFs in Figure 7.
Since the author used the TF index, the spatial resolution of the images obtained by the UAS and the visualization results of the TF index extracted from the images should be given in the paper, but no information about the TF index is given in Figures 6 and 7.
The additional experiment on the effect of different flight altitudes on vegetation indexes (VIs) and textural features (TFs) has been supplemented as recommended (lines 137-139, 328-346, 398-410)
